# Impact of a Prescription Support Tool to Improve Adherence to the Guidelines for the Prescription of Oral Antithrombotics: The Combi-AT Randomized Controlled Trial Using Clinical Vignettes

**DOI:** 10.3390/jcm8111919

**Published:** 2019-11-08

**Authors:** Lorène Zerah, Dominique Bonnet-Zamponi, Agnès Dechartres, Paul Frappé, Marie Hauguel-Moreau, Jean-Philippe Collet, Yann De Rycke, Florence Tubach

**Affiliations:** 1Institut Pierre Louis d’Epidémiologie et de Santé Publique, Institut National de la Santé et de la Recherche Médicale (INSERM), Sorbonne Université, F-75013 Paris, France; 2Observatoire du Médicament, des Dispositifs Médicaux et de l’Innovation Thérapeutique Ile de France (OMEDIT), F-75014 Paris, France; dominique.bonnet-zamponi@inserm.fr; 3Centre de Pharmacoépidémiologie (Cephepi), Département de Santé Publique, Hôpital Pitié Salpêtrière, AP-HP, Institut Pierre Louis d’Epidémiologie et de Santé Publique, Institut National de la Santé et de la Recherche Médicale (INSERM), Sorbonne Université, F-75013, Paris, France; agnes.dechartres@aphp.fr (A.D.); yann.de-rycke@aphp.fr (Y.D.R.);; 4Institut de Recherche en Médecine Générale, F-75005 Paris, France; paul.frappe@univ-st-etienne.fr; 5Département de Cardiologie, Hôpital Pitié Salpêtrière, AP-HP, Sorbonne Université, F-75013 Paris, France; marie.hauguel@aphp.fr (M.H.-M.); jean-philippe.collet@aphp.fr (J.-P.C.)

**Keywords:** antithrombotic, anticoagulant, antiplatelet, combinations, prescription support tool, guidelines

## Abstract

Ensuring the appropriateness of prescriptions of oral antithrombotics (ATs, including antiplatelet and anticoagulant agents) is a crucial safety issue, particularly for patients with multiple chronic conditions. Our main objective was to assess the impact of a prescription support tool, synthesized from international guidelines on oral ATs in adult outpatients, on improving physician adherence to the guidelines for prescription of oral ATs. A web-based, open randomized controlled trial using clinical vignettes was conducted in France from November 2018 to February 2019. General practitioners and cardiologists with outpatient practice were contacted to participate in a web-based survey involving three clinical vignettes illustrating cases of adult outpatients with common neuro-cardiovascular diseases. They were asked to answer four multiple-choice questions related to the number of oral AT(s), drug class, dosage and duration of the prescription. Physicians assigned to the experimental arm had access to the prescription support tool. Physicians assigned to the control arm had no access to the tool. The primary outcome measure was the appropriate prescription of oral ATs (i.e., complied with guidelines in terms of the number, drug class, dosage and duration of prescription). An intent-to-treat analysis was performed using a logistic mixed model with a clinical vignette effect and a physician effect nested in the arm of the trial. Four hundred and forty-one general practitioners and 37 cardiologists were randomized to the experimental (*n* = 238) and to the control arm (*n* = 240), respectively. In the experimental arm, 55.0% of the prescriptions were appropriate versus 29.4% in the control arm (Odds Ratio (OR): 3.61 (2.60 to 5.02)). Access to the first prescription support tool synthesizing the use of oral ATs for outpatients significantly improved the rate of appropriate oral AT prescriptions according to the guidelines.

## 1. Introduction

Antithrombotics (ATs), which include antiplatelet and anticoagulant therapies, are the most frequent drug class implicated in serious and fatal adverse drug events (ADEs) occurring in outpatient settings [1]—among which, 40% to 70% could be preventable [2,3]. AT combinations greatly increase this risk, especially when oral anticoagulation is combined with antiplatelet agents [4,5].

Several studies have reported the inappropriate use of oral ATs in patients with non-valvular atrial fibrillation (concomitant use of drugs increasing the risk of bleeding, insufficient dose of anticoagulant or use of antiplatelet agents instead of anticoagulants) [4,6,7,8]. Regardless of the indication, approximately 15% of patients with AT combinations had inappropriate dual or triple oral AT therapy according to a Canadian survey [9]. However, the appropriateness of the prescribing was limited to the type of drugs combined and did not cover the dosage and duration of prescription. To our knowledge, no data are currently available on the difficulties that physicians have in prescribing these drugs. In addition, no study evaluates the rate of appropriate oral AT prescriptions (in terms of the number of drugs, drug class, dosage and duration of prescription), regardless of the indication, in outpatient settings.

However, ensuring the appropriateness of prescriptions of oral ATs, especially oral AT combinations, is a crucial safety issue, particularly for patients with multiple chronic conditions. We previously showed that international guidelines on oral AT combinations were numerous, although consensual, frequently updated, and that none encompassed all the clinical situations [10]. We hypothesized that this information dispersal was an impediment for physicians to comply with guidelines. Therefore, we designed a prescription support tool based on the systematic synthesis of all relevant guidelines into one single document (Figure 1) [10].

In this current study, we aimed to assess the impact of this prescription support tool on improving physician adherence to the guidelines for prescription of oral ATs (in terms of the number of drugs, drug class, dosage and duration of prescription).

## 2. Material and Methods

The full protocol has been previously published [11]. No change was made to the protocol after the trial began.

### 2.1. Study Design and Setting

An open randomized controlled trial was performed in France by using an online survey targeting physicians managing outpatients with oral ATs. To measure the effect of the prescription support tool, we used clinical vignettes, a method previously shown to be effective to evaluate the quality of care [12,13]. The trial was conducted from November 2018 to February 2019.

### 2.2. Participants

This trial was conducted among French general practitioners and cardiologists identified and contacted by email via physician associations, social networks or a snowball effect. Physicians were asked to answer an exhaustive questionnaire including their characteristics, their degree of self-confidence in prescribing oral AT combinations, their knowledge on the most recent guidelines and where to find them. Physicians with an exclusive hospital practice were not eligible to participate to the trial.

### 2.3. Randomization

Eligible physicians were allocated to the experimental arm (access to the prescription support tool) or to the control arm (no access to the prescription support tool) by randomization (1:1 ratio) with a computer-generated randomization scheme in blocks of 4, stratified by medical specialty (general practitioners or cardiologists). Randomization was centralized through the online survey.

### 2.4. Intervention

In the experimental arm, physicians had access to the prescription support tool and its explanatory guide (Figure 1, Appendix A) [10]. It was developed, from a systematic review of international guidelines published between 2012 and 2018 [10], to help physicians prescribe oral AT combinations for complying with guidelines. This prescription support tool synthesizes, on a double-sided page, selected international guidelines on chronic management (at least 1 month) of oral AT combinations (indication, drugs, dosages and duration) in adults, without considering in-hospital management and bridging therapy (Figure 1) [10]. For the same clinical situation and the same learned society, we selected the latest guidelines to take into account the most recent evidence. To compare recommendations between learned societies, we considered the publication date and the level of evidence of recommendations. We excluded particular clinical situations that require inevitably specialist medical advice: active cancer, autoimmune diseases, hemophilia, HIV, pediatrics and pregnancy. The following pathologies were included in this tool because they are the main causes leading to the prescription of oral ATs (single, dual or triple therapy) in adults [10]: non-valvular atrial fibrillation, coronary artery disease, ischemic stroke, valvular heart disease, peripheral artery disease and venous thromboembolism. For each clinical situation, including one or multiple neuro-cardiovascular condition(s), the number, drug class, dosage and duration of oral ATs to prescribe is given (Figure 1) [10]. At the end, for a same clinical situation, international recommendations were mostly consensual. The recommendations were discordant for only one clinical situation: stable ischemic heart disease with a coronary artery bypass graft. Two guidelines made recommendations about this specific situation. American guidelines [14] recommended a dual antiplatelet therapy for 12 months with a grade IIB (“Benefit ≥ Risks, case by case decision”) and European guidelines [15] recommended a single antiplatelet therapy (with no grade). Considering the level of evidence of the recommendations and their publication date, the scientific committee did not choose between these two guidelines and recommended a “specialist’s opinion” in this case (Figure 1) [10]. Moreover, since the duration of oral AT combinations can be adapted (prolonged or shortened) according to patient-specific risks of ischemia and bleeding, we proposed, in our tool, minimum and maximum durations for oral AT prescriptions (as recommended by learned societies) (Figure 1) [10]. Specifications on which oral ATs should never be combined or are contraindicated are also provided together with the risk stratification tools to use, namely the CHA_2_DS_2_VASc and HAS BLED scores [16,17].

All physicians in both arms had access to the internet and therefore to all available material (guidelines and currently available prescription support tools, etc.).

### 2.5. Clinical Vignettes

Physicians were asked to propose prescriptions for three clinical vignettes illustrating common outpatient clinical situations. For each clinical vignette, multiple-choice questions on the number of oral ATs, drug class of oral ATs, dosage of each oral AT prescribed, and duration of the prescription were proposed. The clinical vignettes were previously created by two physicians (1 cardiologist and 1 internist-geriatrician) in order to cover the common outpatient clinical situations for which the use of oral ATs (none, single, dual or triple therapy) is recommended or needs to be stopped according to the guidelines. The fifth and last question evaluated the self-reported degree of confidence the physician had regarding the appropriateness of his/her prescription on a scale from 0 to 10. A total of 30 clinical vignettes were created (Appendix A). Answers to all clinical vignettes questions can be found in the prescription support tool. An expert committee (1 cardiologist, 1 geriatrician, 1 internist and 2 general practitioners) reviewed all clinical vignettes with the prescription support tool to confirm the likelihood of the clinical vignettes according to clinical practice and their readability. The committee estimated the time needed to complete 3 clinical vignettes at 10 min. To ensure that each clinical vignette was read the same number of times in both arms, we randomly allocated the clinical vignettes to the physicians by using a random-list arm by blocks of 30 vignettes, stratified on the trial arm. After completing the 3 clinical vignettes, physicians of the experimental arm were asked to evaluate, on a scale from 0 and 10, the usefulness of the prescription support tool, how much they would be willing to use it in their practice and if they would recommend its use. They could provide free-text comments. Physicians of the control arm were able to download the prescription support tool once they completed their answers to the 3 clinical vignettes.

### 2.6. Outcomes

The primary outcome measure was the appropriate oral AT prescriptions defined as complying with guidelines in terms of the number of drugs, drug class, dosage and duration of prescription. The correct answers were based on the prescription support tool [10] and validated by the expert committee. Secondary outcomes were: (1) appropriateness of each independent component of the primary outcome measure; (2) self-reported degree of confidence of physicians that their prescription of oral ATs complied with guidelines; and (3) self-reported usefulness of the prescription support tool for physicians assigned to the experimental arm

### 2.7. Data Collection Methods and Data Management

Data from physicians’ answers were automatically integrated in a database for statistical analysis. The data were completely anonymous. In particular, neither the physician’s name nor email address was collected.

### 2.8. Sample Size and Statistical Considerations

The unit of measurement was the prescription (clinical vignettes). Considering that 85% of oral AT prescriptions would comply with guidelines in the control arm [9] and that each physician would complete 3 clinical vignettes, to demonstrate an increase in this proportion up to 90% in the experimental arm, with a power of 80%, an alpha risk of 5% and a two-tailed test, we needed to include a minimum of 230 physicians per arm.

Results were expressed as the mean (standard deviation (SD)) or median (interquartile interval (IQR)) for continuous variables and as numbers (percentages) for categorical variables. For the analysis of the primary outcome, we used an intent-to-treat analysis with a logistic mixed model with a clinical vignette effect and a physician effect nested in the trial arm taking into account that each participant intended to complete 3 clinical vignettes. Missing data were imputed by using multiple imputations with 20 imputed datasets (Appendix A) [18]. Results of the models are expressed as odds ratios (ORs) estimated with Rubin’s rule [19] and 95% confidence intervals (95% CIs). We used the same method to assessed each component of the prescriptions separately and to evaluate the proportion of prescriptions of oral ATs that complied with guidelines by type of recommended prescriptions (no AT required, single therapy (antiplatelet or anticoagulant), dual antiplatelet therapy (2 antiplatelets), dual therapy (single antiplatelet and single anticoagulant) or triple therapy (dual antiplatelet therapy and single anticoagulant). We compared the self-reported degree of confidence of physicians had that their prescription of oral AT combinations complied with guidelines by using a linear mixed model. A sub-group analysis for general practitioners and for cardiologists for the primary outcome was planned a priori.

We performed sensitivity analyses for the handling of missing data with complete-case analysis (per protocol analysis involving physicians who completed at least 1 question for their 3 clinical vignettes and taking into account only the observed answers) and modified intent-to-treat analysis (involving physicians who completed at least 1 question for their 3 clinical vignettes and imputing missing data). All analyses involved use of R v3.5.2 (www.cran.r-project.org).

## 3. Results

### 3.1. Population and Clinical Vignette Characteristics

Among the 680 physicians who connected to the website, 478 were randomized, including 441 general practitioners (92%) and 37 cardiologists (8%), between 5 November 2018 and 14 February 2019. In total, 238 were randomized to the experimental arm and 240 were randomized to the control arm (Figure 2).

Baseline characteristics of randomized physicians are presented in Table 1.

Many physicians had to deal with prescriptions of oral AT combinations in their daily practice; indeed, 38% (*n* = 183) reported that at least 5% of their patients took an oral AT combination. Overall, 71% (*n* = 336) did not feel comfortable with these prescriptions (not comfortable at all or rather uncomfortable) and 43% (*n* = 205) declared not knowing the guidelines for oral AT combinations or where to find them. Cardiologists felt more comfortable with these prescriptions (76% of cardiologists vs. 26% of general practitioners felt rather comfortable or very comfortable with prescribing oral AT combinations) and more often knew where to find the guidelines (92% vs. 54%).

Ninety physicians were randomized but did not complete any of their three allocated clinical vignettes (Figure 2). Their baseline characteristics did not differ from those who completed at least one question for their three allocated clinical vignettes, except for two points: noncompleters more often belonged to the experimental arm than completers (80% vs. 43%) and fewer did not know the most recent guidelines on oral AT combinations or where to find them (32% vs. 45%) (Appendix A).

The distribution of the 30 clinical vignettes by trial arm is in Appendix A.

### 3.2. Primary Outcome

In all, 55.0% (*n* = 393) of oral AT prescriptions were fully appropriate in the experimental arm versus 29.4% (*n* = 212) in the control arm: OR: 3.61 (95% CI 2.60 to 5.02) (Table 2). The most common prescriptions that did not comply with guidelines in the control arm were prescriptions of an AT when no AT was recommended, wrong type of AT when one AT was recommended (prescription of antiplatelet instead of anticoagulant and vice versa), inappropriate doses of anticoagulant and duration of combinations of oral ATs (Table 2). When a single anticoagulant was recommended, anticoagulant doses were either under-dosed for the prescription of direct oral anticoagulants for non-valvular atrial fibrillation (87%) or over-dosed for the prescription of a vitamin K antagonist for valvular disease (100%). When dual therapy was recommended, anticoagulant doses were over-dosed in 96% of cases. Combinations of oral ATs were usually prescribed longer than recommended (100% of all prescriptions of triple therapies, 93% of all prescriptions of dual therapies and 76% of all prescriptions of dual antiplatelet therapies).

### 3.3. Secondary Outcomes

#### 3.3.1. Appropriate Prescriptions of Oral ATs by Type of Prescription

Physicians assigned to the experimental arm prescribed oral ATs more frequently in accordance with guidelines than did physicians assigned to the control arm when a single antiplatelet treatment was required (65% vs. 42%, OR: 2.89 (1.35 to 6.18), when a single anticoagulant treatment was required (47% vs. 29%, OR: 2.34 (1.47 to 3.72) and when a dual antiplatelet treatment was required (61% vs. 40%, OR: 2.46 (1.42 to 4.26) (Table 2). There was no significant difference when no treatment was required or when a dual or triple therapy was required (Table 2).

#### 3.3.2. Individual Components of Oral AT Prescription

When considering all clinical vignettes, each component of the prescription was significantly improved when physicians had access to the prescription support tool, with ORs ranging from 2 to 4 (Table 2). Especially, access to the tool improved the compliance with guidelines in terms of anticoagulants’ doses (single anticoagulant: 84% vs. 68%, OR = 3.04 (1.27 to 7.30); dual therapy: 84% vs. 49%, OR = 6.31 (2.06 to 19.37)) and of duration of ATs combinations (dual antiplatelet: 72% vs. 53%, OR = 2.63 (1.46 to 4.72); dual therapy: 70% vs. 42%, OR = 3.52 (1.65 to 7.51); triple therapy: 69% vs. 26%, OR = 6.54 (1.34 to 31.85)) (Table 2). In contrast, the use of the prescription support tool did not significantly improve the prescription of antiplatelet doses, the prescription duration of single anticoagulant or the prescription of triple therapies (with the exception of duration) (Table 2).

#### 3.3.3. Self-Confidence of Physicians

With the prescription support tool, physicians felt much more confident prescribing according to the guidelines when using the prescription support tool as compared to without (median degree of confidence (/10) 9 (IQR 6 to 9) vs. 6 (4 to 8) (OR 4.40 (3.16 to 6.14, *p* < 0.001) (Table 2). These data were significant for all types of prescriptions, with 2 exceptions: triple therapy or when no treatment was required (Table 2).

#### 3.3.4. Evaluation of the Prescription Support Tool

In all, 60% (*n* = 144) of physicians in the experimental arm evaluated the prescription support tool. They found it useful (median score (/10) 9 (IQR 8 to 10)), would be ready to use it (median score (/10) 9 (8 to 10)) and would recommend its use (median score (/10) 9 (8 to 10)) (Table 3). The scores were slightly lower for the clarity of the tool (median score (/10) 7 (5 to 9)) and its operation (median score (/10) 7 (5 to 9)). Overall, 35% (*n* = 51) of physicians provided free-text comments, including the creation of a digital tool usable on a computer or smartphone for 12% of them (Table 3).

### 3.4. Sensitivity Analyses

Sensitivity analyses performed regarding the handling of missing data gave results similar to the main analysis (Appendix A).

## 4. Discussion

### 4.1. Key Findings

Among a sample of French physicians with clinical practice in outpatient settings, two-thirds were uncomfortable with prescribing oral AT combinations and almost half did not know where to find the recommendations. When evaluating clinical vignettes representing common clinical situations, only 29.4% of oral AT prescriptions were appropriate in the control arm (reflecting clinical practice with currently available prescription support tools), and the median self-reported degree of confidence these physicians had regarding the appropriateness of their prescriptions was 6 (/10) (IQR 4 to 8). The use of a dedicated prescription support tool synthesizing the guidelines on oral ATs led to significant improvement in the proportion of appropriate oral AT prescriptions (55% in the experimental arm; OR: 3.61 (95% CI 2.60 to 5.02)). Moreover, physicians felt much more confident prescribing according to the guidelines when they use the prescription support tool as compared to those who did not (median degree of confidence (/10): 9 (IQR 6 to 9), *p* < 0.001). To our knowledge, this is the first study reporting such results.

### 4.2. Comparison with Other Studies

Overall, one-third of oral AT prescriptions were fully appropriate without access to the prescription support tool. This proportion is much lower than the 85% estimated rate from literature data [9] based on a single real-life Canadian study, where the dosage and the duration were not taken into account. However, inappropriate doses of oral anticoagulant and an inappropriate duration of oral AT combinations were the main errors of physicians in the control arm, which is consistent with the literature [4,6,7,8]. Therefore, failure to consider all components of oral AT prescriptions in the previous study [9] may have overestimated the appropriate oral AT prescription rate. To our knowledge, Combi-AT is the first study evaluating the rate of appropriate oral AT prescriptions considering all components of the prescription (i.e., the number of drugs, drug class, dosage and duration of prescription), regardless of the indication, in outpatient setting.

Frequent updating of recommendations [10] do not solely explain the high rates of suboptimal prescriptions. The indications for single antiplatelet therapy or single anticoagulant therapy did not change during the past few years [10], but only 42% of clinical vignettes with an indication for single antiplatelet therapy and 29.4% of clinical vignettes with an indication for single anticoagulant therapy were prescribed according to the guidelines in the control arm. Our randomized study demonstrates that the dissemination of the guidelines alone is not sufficient to optimize performance [20], and that access to the relevant guidelines remains a major issue. Providing physicians with unique and ergonomic access to all recommendations seems necessary to narrow the gap between clinical practice and guidelines recommendations.

Our results also outline that many patients with atrial fibrillation who have an indication for chronic anticoagulant remain undertreated with an elevated risk of stroke [7,21]. The reasons are multiple including physician education and the fear of side effects, especially bleeding events [22]. Our prescription support tool provides both a reminder of the recommendations with respect to indication and adequate anticoagulant doses but also risk stratification by a systematic use of CHA_2_DS_2_VASc and HAS BLED algorithms [10,16,17]. This may have facilitated the process and made the physicians feel more comfortable using a step-by-step decision-making process, which is especially useful when there are multiple pathologies.

### 4.3. Strengths and Limitations

This prescription support tool is unique, including all national and international guidelines for the management of oral AT prescriptions in adult outpatients [10]. To evaluate the prescription support tool, we used an original and validated method based on clinical vignettes [12,13]. The selected clinical vignettes were reviewed by an expert committee to ensure the likelihood of the real-life clinical situations. This first step was essential to assessing physicians’ difficulties in accessing recommendations and to developing new and effective strategies in the future. This tool could be used by physicians, pharmacists, nurses and could be an aid to a structured medication review.

A large number of physicians who are used to prescribing oral ATs in their daily practice volunteered and were enrolled but a selection bias cannot be excluded. Conversely, non-access to the prescription support tool among the physicians assigned to the control arm could not be guaranteed, as was its use in the experimental arm. However, these potential biases would have reduced the impact of our tool.

Only a few cardiologists participated in the study despite multiple incentives and so reliable subgroup analyses by type of physicians could not be performed. Nevertheless, the target users of our tool are mainly general practitioners for several reasons. First, because comorbid conditions are very common in patients with cardiovascular disease [23], and patients with oral ATs indications are usually followed by a general practitioner and not just a specialist. The general practitioner has a coordinating role among all health professionals. If cardiologists or neurologists generally make the first oral AT prescription during one hospitalization (drugs and duration of the first treatment phase), the general practitioner must ensure the proper implementation of oral AT management in outpatient settings. By recalling the chronological stages of the therapeutic regimen, our tool may help general practitioners to better coordinate with health professionals (e.g., anticipation of specialist appointment) and to adapt oral ATs when access to cardiologists or neurologists is difficult. The average waiting time for a consultation with a general practitioner in France was 6 days in 2018 vs. 50 days for a cardiologist [24]. Second, cardiologists are not the primary prescribers of oral ATs in the elderly. However, older people are the main consumers of oral ATs and the most at risk population for inappropriate oral AT prescriptions and adverse drug events [25]. In a recent French observational study focusing on the use of direct oral anticoagulants in older outpatients (i.e aged 75 years and older; *N* = 19,798 outpatients) [26], Barben et al. showed that first prescriptions came primarily from specialists other than cardiologists (45.4%) or general practitioners (44.4%), and least often from cardiologists (10.2%). Regarding refill prescriptions, the prescribers for the vast majority were general practitioners (94.3%); other specialists made up 3.8% of prescribers and cardiologists only 1.8%. Lastly, we assumed that cardiologists were more familiar with the guidelines than general practitioners, which was confirmed by our findings (they felt more comfortable and they more often knew where to find the guidelines). Consequently, general practitioners were particularly enthusiastic about our tool (free-text comments), stating that they found it useful, that they would be ready to use it and that they would recommend its use. Whether the findings of this French study are generalizable is a pending issue.

More physicians in the experimental arm did not respond to their clinical vignettes than in the control arm. There was no impact of missing data, and sensitivity analyses gave similar results.

Finally, even with access to the prescription support tool, only 55% of the prescriptions were fully appropriate, which could reflect the non-optimal format of the tool (median operation score (/10): 7 (IQR 5 to 9)) and/or the fact that physicians did not make complete use of it because there was no real clinical issue or because they thought they knew the answer. In fact, we do not know how many clinical vignettes have been completed using the tool. Fifty-one physicians in the experimental arm provided free-text comments about the tool. Twenty-four physicians found that the tool was difficult to read due to too small writing and too many abbreviations. By improving the format of the tool, and in particular its readability and ergonomics, we can expect a greater use and a better rate of appropriate oral AT prescriptions.

### 4.4. Perspectives

A paper format of the tool is not suitable for frequent updating of guidelines on ATs, which occurs approximately every 5 years [10]. This delay is even shorter when considering the European Society of Cardiology guidelines, where overlap is frequent and leads to changes every year [15,27]. An interactive web tool, based on this prescription support tool, is being created to fulfil these constraints, improve readability and ergonomics and to assess whether such an approach may have an impact on patient’s clinical outcomes (ischemic and hemorrhagic risk).

## 5. Conclusions

Oral AT prescriptions were three times more frequently appropriate when physicians had access to a prescription support tool than without. It should therefore interest all physicians caring for adults with neuro-cardiovascular pathologies requiring oral ATs (single, dual and triple) in order to prescribe with greater safety and confidence.

## Figures and Tables

**Figure 1 jcm-08-01919-f001:**
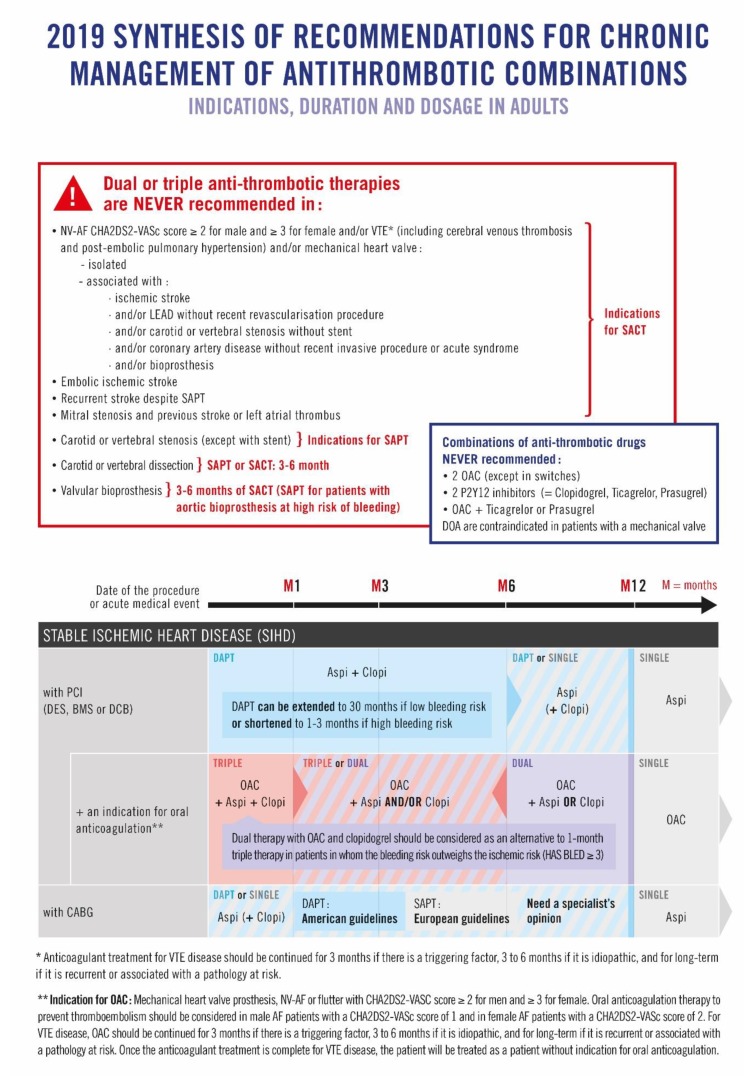
The 2019 synthesis of recommendations for the chronic management of antithrombotic combinations.

**Figure 2 jcm-08-01919-f002:**
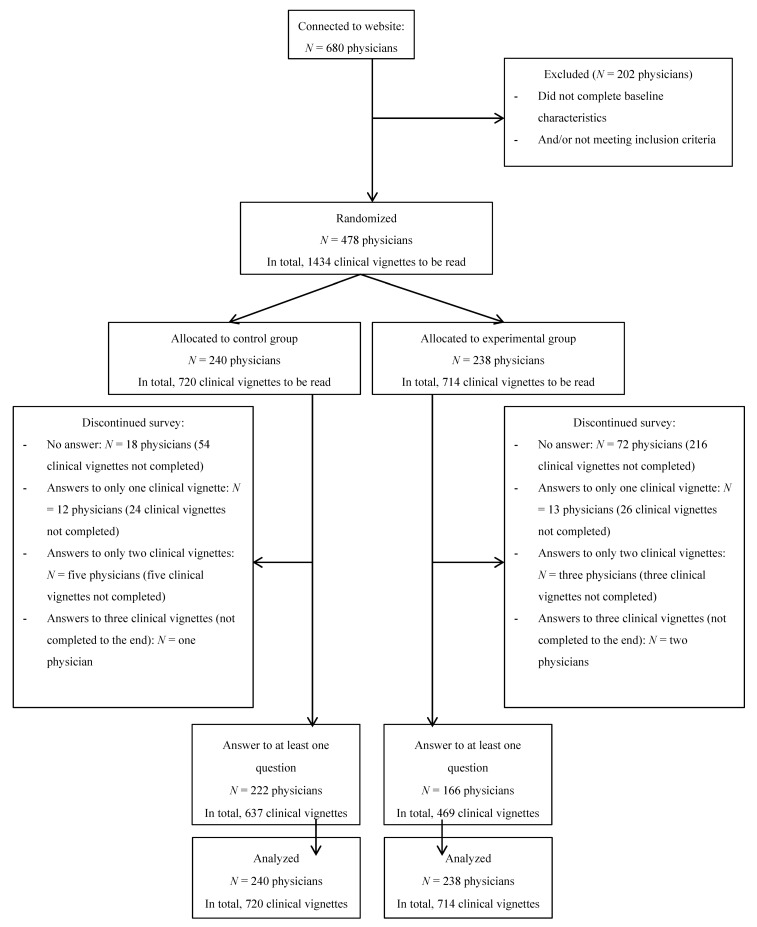
Flow chart.

**Table 1 jcm-08-01919-t001:** Baseline characteristics of randomized physicians. Values are numbers (percentages) unless stated otherwise.

	Experimental Group	Control Group	*p* Value
	(*N* = 238)	(*N* = 240)	
**Sex**			
Male	112 (47)	106 (44)	0.6
Female	126 (53)	134 (56)	
**Mean (SD) age (years)**	44 (14)	42 (13)	0.2
**Specialty:**			
GP	220 (92)	221 (92)	0.9
Cardiologists	18 (8)	19 (8)	
**Years since graduation**			
≤1	7 (3)	7 (3)	
2 to 5	54 (23)	57 (24)	
6 to 10	52 (22)	57 (24)	0.7
11 to 20	36 (15)	43 (18)	
≥21	89 (37)	76 (32)	
**Percentage of patients taking oral AT combinations**			
≤5%	146 (61)	149 (62)	
6 to 10%	62 (26)	63 (26)	0.9
11 to 20%	25 (11)	21 (9)	
≥21%	5 (2)	7 (3)	
**Whether physicians feel comfortable or not with the management of oral AT combinations**			
Not comfortable at all	47 (20)	42 (18)	
Rather uncomfortable	122 (51)	125 (52)	0.5
Rather comfortable	63 (26)	68 (28)	
Very comfortable	6 (3)	5 (2)	
**Guidelines about oral AT combinations: do you know them and where to find them?**			
No and I do not know where to find them	96 (40)	109 (45)	
No, but I know where to find them	104 (34)	92 (37)	0.9
Yes, and I know where to find them	38 (16)	39 (16)	

Abbreviations: AT: antithrombotic; GP: general practitioner.

**Table 2 jcm-08-01919-t002:** Primary and secondary outcomes: compliance with guidelines for oral AT prescriptions by trial arm. Values are numbers (percentages) unless stated otherwise.

	Experimental Arm	Control Arm	OR (95%CI)	*p*
	*N* = 238	*N* = 240		
**All allocated clinical vignettes**				
	***n* = 714 CV**	***n* = 720 CV**		
**PRIMARY OUTCOME**	393 (55.0)	212 (29.4)	3.61 (2.60 to 5.02)	*p* < 0.001
**Fully appropriate prescription**				
**Each component of the prescription**				
Number of oral ATs	567 (79.4)	437 (60.7)	2.76 (2.03 to 3.76)	*p* < 0.001
Type of oral ATs	502 (70.4)	382 (53.1)	2.27 (1.69 to 3.06)	*p* < 0.001
Dosage of oral ATs	450 (89.6)	282 (73.8)	4.13 (2.24 to 7.61)	*p* < 0.001
Duration of the prescription	561 (78.6)	444 (61.7)	2.56 (1.87 to 3.50)	*p* < 0.001
**Median (IQR) self-reported degree of confidence (/10)**	9 (6 to 9)	6 (4 to 8)	4.40 (3.16 to 6.14)	*p* < 0.001
**Clinical vignettes with recommendation of no AT treatment**				
	***n* = 24 CV**	***n* = 24 CV**		
**Fully appropriate prescription**	15 (60.4)	7 (29.4)	3.61 (0.96 to 13.51)	*p* = 0.06
**Number of oral ATs**	15 (60.4)	7 (29.4)	3.61 (0.96 to 13.51)	*p* = 0.06
**Median (IQR) self-reported degree of confidence (/10)**	6 (5 to 8)	6 (5 to 8)	1.96 (0.51 to 7.62)	*p* = 0.32
**Clinical vignettes with recommendation of single antiplatelet therapy**				
	***n* = 117 CV**	***n* = 119 CV**		
**Fully appropriate prescription**	76 (65.3)	50 (42.0)	2.89 (1.35 to 6.18)	*p* = 0.007
**Each component of the prescription**				
Number of oral ATs	102 (86.7)	84 (70.3)	3.06 (1.29 to 7.25)	*p* = 0.01
Type of oral ATs	88 (74.4)	57 (48.0)	3.36 (1.59 to 7.10)	*p* = 0.001
Dosage of oral ATs	85 (96.6)	55 (96.5)	1.37 (0.17 to 11.11)	*p* = 0.76
Duration of the prescription	94 (80.5)	76 (64.2)	2.39 (1.04 to 5.53)	*p* = 0.04
**Median (IQR) self-reported degree of confidence (/10)**	7 (5 to 8)	6 (4 to 7)	3.11 (1.49 to 6.49)	*p* = 0.003
**Clinical vignettes with recommendation of single anticoagulant therapy**				
	***n* = 263 CV**	***n* = 266 CV**		
**Fully appropriate prescription**	124 (47.0)	78 (29.4)	2.34 (1.47 to 3.72)	*p* < 0.001
**Each component of the prescription**				
Number of oral ATs	189 (71.9)	155 (58.4)	1.95 (1.21 to 3.14)	*p* = 0.006
Type of oral ATs	168 (63.9)	142 (53.2)	1.73 (1.06 to 2.85)	*p* = 0.03
Dosage of oral ATs	141 (83.9)	97 (68.3)	3.04 (1.27 to 7.30)	*p* = 0.01
Duration of the prescription	230 (87.6)	221 (83.1)	1.54 (0.46 to 5.22)	*p* = 0.48
**Median (IQR) self-reported degree of confidence (/10)**	7 (6 to 8)	6 (4 to 8)	3.76 (2.36 to 5.97)	*p* < 0.001
**Clinical vignettes with recommendation of dual antiplatelet therapy**				
	***n* = 168 CV**	***n* = 168 CV**		
**Fully appropriate prescription**	102 (61.0)	68 (40.4)	2.46 (1.42 to 4.26)	*p* = 0.001
**Each component of the prescription**				
Number of oral ATs	140 (83.2)	114 (68.1)	2.55 (1.38 to 4.70)	*p* = 0.003
Type of oral ATs	125 (74.5)	104 (62.1)	1.86 (1.05 to 3.29)	*p* = 0.03
Dosage of oral ATs	120 (96)	90 (86.5)	4.22 (0.01 to 2000)	*p* = 0.67
Duration of the prescription	121 (72.2)	90 (53.5)	2.63 (1.46 to 4.72)	*p* = 0.001
**Median (IQR) self-reported degree of confidence (/10)**	8 (7 to 9)	6 (4 to 8)	5.20 (3.03 to 8.95)	*p* < 0.001
**Clinical vignettes with recommendation of dual therapy**				
	***n* = 118 CV**	***n* = 119 CV**		
**Fully appropriate prescription**	53 (44.9)	8 (6.8)	26.2 (0.25 –2716)	*p* = 0.16
**Each component of the prescription**				
Number of oral ATs	92 (78.1)	63 (53.1)	3.29 (1.60 to 6.79)	*p* = 0.001
Type of oral ATs	80 (67.4)	61 (51.3)	2.08 (1.03 to 4.20)	*p* = 0.04
Dosage of oral ATs	67 (83.7)	30 (49.2)	6.31 (2.06 to 19.37)	*p* = 0.002
Duration of the prescription	83 (70.0)	50 (42.2)	3.52 (1.65 to 7.51)	*p* = 0.001
**Median (IQR) self-reported degree of confidence (/10)**	7 (6 to 9)	5 (4 to 7)	6.09 (3.02 to 12.29)	*p* < 0.001
**Clinical vignettes with recommendation of triple therapy**				
	***n* = 24 CV**	***n* = 24 CV**		
**Fully appropriate prescription**	0 (0)	0 (0)		
**Each component of the prescription**				
Number of oral ATs	12 (51.9)	13 (53.1)	0.95 (0.25 to 3.57)	*p* = 0.94
Type of oral ATs	13 (54.2)	12 (48.1)	1.28 (0.34 to 4.74)	*p* = 0.71
Dosage of oral ATs	0 (0)	1 (4.2)		
Duration of the prescription	17 (69.4)	6 (26.2)	6.54 (1.34 to 31.85)	*p* = 0.02
**Median (IQR) self-reported degree of confidence (/10)**	5 (2 to 7)	5 (3 to 6)	3.53 (0.62 to 20.00)	*p* = 0.15

Abbreviations: AT: antithrombotic; OR: odds ratio; CI: confidence interval; IQR: interquartile range. Analysis involved multiple imputations (20 imputed datasets, seed = 2019) with a logistic mixed model for categorical variables and a linear mixed model for quantitative variables. Dosage of oral ATs: the % is given for physicians who chose the right type of oral ATs to the previous question.

**Table 3 jcm-08-01919-t003:** Evaluation of the prescription support tool reported by 144 physicians randomized to the experimental group (scores out of 10). Values are medians (IQR).

Experimental Group	
This document helped me to prescribe antithrombotics for clinical vignettes	**9** (7 to 10)
This document has modified the answers that I would have made spontaneously	**7** (5 to 9)
This document is clear	**7** (5 to 9)
This document is operational	**7** (5 to 9)
This document is useful for practice	**9** (8 to 10)
I would be ready to use this tool	**9** (8 to 10)
I would recommend the use of this tool	**9** (8 to 10)
**Free Comments:** ● No comments: *N* = 93 (65%) ● Comments: *N* = 51 (35%) ○ Asked for a digital version of the tool: *N* = 17 (12%) ○ Thanked and congratulated for the tool: *N* = 12 (8%) ○ Others (hard to read it, writing too small, too many abbreviations): *N* = 24 (17%)	

Abbreviations: IQR: interquartile range.

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
