# Peer review of "Impact of a Prescription Support Tool to Improve Adherence to the Guidelines for the Prescription of Oral Antithrombotics: The Combi-AT Randomized Controlled Trial Using Clinical Vignettes"

_jcm, 2019, doi:10.3390/jcm8111919_

Round 1
Reviewer 1 Report
In this paper a collaborative group of epidemiologists, statisticians and internal medicine doctors present a study on the therapeutic attitude of French general practitioners concerning prescription of antithrombotic therapies, with special regard to choice, dosage, duration and various drug associations. The study is an open controlled randomized trial including a support tool supplied to the participants, that summarizes a number of dependable guidelines on various aspects of antithrombotic therapy (AT). The participants afterwards receive three "vignettes" about different therapeutic situations of AT to which they are asked to replay with a clear therapeutic choice. A control group of general practitioners do not receive the above basic material but only the clinical "vignettes", to which they have to answer.
Results show that correct answers are almost 3 times more frequent in the experimental group versus the control group. Thus, the supporting tool that supplies information from the guidelines is effective in improving the performance of the general practitioners.
The study is original and interesting. However, I have a few questions.
1) The synthesis of a number of good guidelines is a difficult task. How did Authors reconcile the unavoidable contradictions? How could they avoid personal preferences and biases?
2) The much better performance of the group of doctors who had been informed with the guidelines is an expected result. It could be rather unexpected that the performance of the control group wa so poor!
3) But, what is more, also the performance of the "experimental" group, fed with the best information, was only 55%! Authors should discuss not only the differences between the two groups, but also the actual values of each group. It is unconfortable to learn that about 45% of the general practitioners failed to give correct answers.
4) In fact, it is natural that good information improves the medical performance, but it is discouraging that, despite good information, almost 50% of the doctors gave incorrect answers. I am waiting for discussion and proposals about this worrying and unsettling question.
Author Response
We thank the reviewer 1 for her/his helpful comments.
1/ The synthesis of a number of good guidelines is a difficult task. How did Authors reconcile the unavoidable contradictions? How could they avoid personal preferences and biases?
The synthesis of guidelines was a difficult task because clinical situations concerned by oral antithrombotic (AT) prescriptions were numerous, and the recommendations covering all these situations were contained in 70 guidelines (Zerah et al, Plos one 2019 [10]). For the same clinical situation and the same learned society, we selected the latest guidelines to take into account the most recent evidence. To compare recommendations between learned societies, we considered the publication date and the level of evidence of recommendations.
At the end, for a same clinical situation, international recommendations were mostly consensual. The recommendations were discordant for only one clinical situation: stable ischemic heart disease with a coronary artery bypass graft. Two guidelines made recommendations about this specific situation. American guidelines (Levine et al, 2016 [14]) recommended a dual antiplatelet therapy for 12 months with a grade IIB (“it may be reasonable”); European guidelines (Valgimigli et al, 2017 [15]) recommended a single antiplatelet therapy (no grade). Considering the level of evidence of recommendations and their publication date, the scientific committee did not choose between these two guidelines and recommended a “specialist’s opinion” in this case (Figure 1). Moreover, since the duration of oral AT combinations can be adapted (prolonged or shortened) according to patient-specific risks of ischemia and bleeding, we proposed, in our tool, minimum and maximum durations for oral AT prescriptions (as recommended by learned societies) (Figure 1).We added the precision that the guidelines were “numerous although consensual” in the introduction (lines 64 - 64) and specified how was resolved the single clinical situation for which guidelines were discordant in the intervention paragraph (lines 105 – 108 and 115 - 126)
2/ The much better performance of the group of doctors who had been informed with the guidelines is an expected result. It could be rather unexpected that the performance of the control group was so poor!
We agree that this result was really surprising: only one third of oral AT prescriptions were fully appropriate without access to the prescription support-tool.
This proportion is much lower than the 85% estimated rate from the literature data [Hamilton et al, 2016 [9]] based on a single Canadian real life study. In this study, dosage and duration were not taken into account. However, inappropriate doses of oral anticoagulant and inappropriate duration of oral AT combinations were the main errors of physicians in the control arm, which is consistent with the literature. Therefore, failure to consider all components of oral AT prescriptions in the previous study [Hamilton et al, 2016 [9]] may have overestimated the appropriate oral AT prescription rate. In our study, 49% of oral AT prescriptions in the control group were appropriate considering only the number of drugs and the drug class.
To our knowledge, Combi-AT is the first study evaluating the rate of appropriate oral AT prescriptions considering all components of prescription (ie number of drugs, drug class, dosage and duration of prescription), regardless of the indication, in outpatient setting.
We have specified all these elements in the discussion (lines 294 - 300)
3-4/ But, what is more, also the performance of the "experimental" group, fed with the best information, was only 55%! Authors should discuss not only the differences between the two groups, but also the actual values of each group. It is uncomfortable to learn that about 45% of the general practitioners failed to give correct answers. In fact, it is natural that good information improves the medical performance, but it is discouraging that, despite good information, almost 50% of the doctors gave incorrect answers. I am waiting for discussion and proposals about this worrying and unsetting question.
We previously discussed (question 2, lines 294 – 300) the low performance of physicians in the control arm. Although the performance of physicians in the experimental arm is much better, it is true that it is far from optimal. Even with access to the prescription support-tool, only 55% of the prescriptions were fully appropriate.
As stated in the discussion, it could reflect: the non-optimal format of the tool (median operation score [/10]: 7 [IQR 5 to 9]) and/or the fact that physicians did not make a complete effort to use it because there was no real clinical issue or because they thought they knew the answer (lines 359- 362).
In fact, we do not know how many clinical vignettes have been completed using the tool. Fifty-one physicians in the experimental arm provided free-text comments about the tool. Twenty-four physicians found that the tool was difficult to read due to too small writing and too many abbreviations. By improving the format of the tool, and in particular its readability and ergonomics, we can expect a greater use and a better rate of appropriate oral AT prescriptions (lines 362 – 367).
The digital version of the tool, currently being developed and designed with general practitioners, should address these issues, as well as those related to frequent updates (lines 372 - 374). This first step was essential to assess physicians’ difficulties in accessing recommendations and to develop new and effective strategies in the future (lines 322 - 324).
Reviewer 2 Report
Summary:
The submitted manuscript is an investigation on the impact of support-tools in improving physician adherence to guidelines for prescription of oral antithrombotics (AT).
The authors designed a clinical trial, COMBI-AT, with the primary outcome of interest being the appropriate oral ATs prescription defined as complying with guidelines in terms of number of drugs, drug class, dosage and duration of prescription.
In an open-labeled randomized controlled trial, 478 general physician (GP) and cardiologists were enrolled to the experimental and control arms to complete the designed clinical vignettes.
In an intention-to-treat analysis, 55.0% of the prescriptions in experimental arm were appropriate versus 29.4% in the control arm (OR: 3.61 (2.60 to 5.02)).
The authors concluded that “AT prescriptions were three times more frequently appropriate when physicians had access to a prescription support-tool than without. It should therefore interest all physicians caring for adults with neuro-cardiovascular pathologies requiring oral ATs (single, dual and triple) in order to prescribe with greater safety and confidence.”
Strength:
With the current spars, various and ever-changing guidelines on AT prescriptions, there is a undeniable need for some clinical tools to guide the physicians and this study the first one of its kind to evaluate the impact of support-tools in prescribing AT medications.
Detailed comments:
Please revise the sentence “Our main objective was to assess the impact of a prescription support-tool, synthesizing international guidelines on oral ATs in adult outpatients, to improve adherence to the guidelines for the prescription of oral ATs.” as following; “Our main objective was to assess the impact of a prescription support-tool, synthesized from international guidelines on oral ATs in adult outpatients, on improving the physician adherence to the guidelines for prescription of oral ATs.”Please consider revising the first sentence of second paragraph (line 51-54) for a better readability.
On main concern is that general physicians constitute the majority of the study population. While this group are at the front-line of patient management, dealing with AT prescription is usually out of their scope of practice especially in AT initiation and termination, where the cardiologist is the responsible physician to establish the indication and duration. Therefore, cardiologist constituting about 16% of study population is a major limitation and assessing the impact of a support-tool in a group that would not be the target users is questionable.
Another important issue is loss of more than 30% of study population in the experimental arm. The authors decided to run an intent-to-treat (ITT) analysis by imputing data for the missing values. While the study is an outcome analysis not a prognosis analysis, a per-protocol (PP) approach could be justified and it is not clear why the authors decided to run an ITT. It would be of interest to evaluate the difference between the ITT and PP analysis and even if authors argue that a per protocol analysis would introduce bias, they should justify their choice in the text. Regarding the designed clinical vignettes, the authors indicated that “An expert committee (1 cardiologist, 1 geriatrician, 1 internist and 2 general practitioners) reviewed all clinical vignettes with the prescription support-tool (external validation) to confirm the likelihood of the clinical vignettes according to clinical practice and their readability.” However, the external validation seems to be more a consensus-based evaluation without any statistical analysis. In my opinion, there is a need for inter and intra-observer analysis to evaluate the reliability and reproducibility of the outcomes of the designed support-tool.
Please consider revising the secondary outcomes wording (line 143-146) for a more precise text. I suggest “Secondary outcomes were: (1) appropriateness of each independent component of the primary outcome measure; (2) self-reported degree of confidence of physicians that their prescription of oral ATs complied with guidelines; and (3) self-reported usefulness of the prescription support-tool for physicians assigned to the experimental arm.
Please provide the p-values for table 1 where indicated. The group differences are not clear.
Author Response
We thank the reviewer 2 for her/his helpful comments.
Please revise the sentence: “Our main objective was to assess the impact of a prescription support-tool, synthesizing international guidelines on oral ATs in adult outpatients, to improve adherence to the guidelines for the prescription of oral ATs” as following; “Our main objective was to assess the impact of a prescription support-tool, synthesized from international guidelines on oral ATs in adult outpatients, on improving the physician adherence to the guidelines for prescription of oral ATs”.
The modification was made in the abstract (lines 23 – 25) and in the introduction paragraph (line 75).
Please consider revising the first sentence of second paragraph (line 51-54) for a better readability.
We revised the sentence: “A recent study, based on the French national healthcare databases, found several situations of inappropriate use of direct oral anticoagulant initiation in patients with non-valvular atrial fibrillation (NV-AF), including concomitant use of drugs increasing the risk of bleeding (1 in 3 new users) and potential underdosing [6].” as following “Several studies have reported the inappropriate use of oral ATs in patients with non-valvular atrial fibrillation (concomitant use of drugs increasing the risk of bleeding, insufficient dose of anticoagulant or use of antiplatelet agents instead of anticoagulants)” and we have added new citations (Xian et al, JAMA, 2017).
Introduction paragraph (lines 51 - 53).
On main concern is that general physicians constitute the majority of the study population. While this group are at the front-line of patient management, dealing with AT prescription is usually out of their scope of practice especially in AT initiation and termination, where the cardiologist is the responsible physician to establish the indication and duration. Therefore, cardiologist constituting about 16% of study population is a major limitation and assessing the impact of a support-tool in a group that would not be the target users is questionable.
The target users of the tool are mainly general practitioners for several reasons:
1° First, because comorbid conditions are very common in patients with cardiovascular disease [23], patients with oral ATs indications are usually followed by a general practitioner and not just a specialist. The general practitioner has a coordinating role among all health professionals. If cardiologists or neurologists generally make the first oral AT prescription during one hospitalization (drugs and duration of the first treatment phase), the general practitioner must ensure the proper implementation of oral AT management in outpatient settings. By recalling the chronological stages of the therapeutic regimen, our tool may help general practitioners to better coordinate with health professionals (e.g., anticipation of specialist appointment) and to adapt oral ATs when access to cardiologists or neurologists is difficult. The average waiting time for a consultation with a general practitioner in France was 6 days in 2018 vs 50 days for a cardiologist [24].
2° Cardiologists are not the primary prescribers of oral ATs in the elderly. Yet, older people are the main consumers of oral ATs and the most at risk population for inappropriate oral AT prescriptions and adverse drug events [25]. In a recent French observational study focusing on the use of direct oral anticoagulants in older outpatients (i.e aged 75 years and older; N = 19 798 outpatients) [26], Barben et al showed that first prescriptions came primarily from specialists other than cardiologists (45.4%) or general practitioners (44.4%), and least often from cardiologists (10.2%). Regarding refill prescriptions, the prescribers were for the vast majority general practitioners (94.3%); other specialists made up 3.8% of prescribers and cardiologists only 1.8%.
3° We assumed that cardiologists were more familiar with the guidelines than general practitioners, which was confirmed by our findings (they felt more comfortable and they more often knew where to find the guidelines).
Consequently, general practitioners were particularly enthusiastic about our tool (free text), they found it useful, they would be ready to use it and they would recommend its use.
We have specified all these elements in the discussion (lines 332 - 355).
Another important issue is loss of more than 30% of study population in the experimental arm. The authors decided to run an intent-to-treat (ITT) analysis by imputing data for the missing values. While the study is an outcome analysis not a prognosis analysis, a per-protocol (PP) approach could be justified and it is not clear why the authors decided to run an ITT. It would be of interest to evaluate the difference between the ITT and PP analysis and even if authors argue that a per protocol analysis would introduce bias, they should justify their choice in the text
This is a superiority trial, and we made an ITT analysis as principal analysis as recommended. Actually, the ITT analysis aims at preserving the benefit of randomisation (avoiding selection bias due to the exclusion of participants) and is a more conservative approach than PP analysis. In superiority trials, any over-estimation of the effect needs to be avoided. With respect to prevention of type I error, it is better to choose a method which under-estimates the effect (conservative approach as the ITT approach) than a method which might over-estimate it (as the PP approach).
However, as specified in the statistical considerations paragraph, we performed sensitivity analyses with different ways of handling missing data and a per protocol analysis (complete-case analysis). Complete-case analysis (per protocol analysis) gave results similar to the main analysis (Supplementary file 7 and 8).
We specified that complete-case analysis was per protocol analysis in the statistical considerations paragraph (line 187)
Regarding the designed clinical vignettes, the authors indicated that “An expert committee (1 cardiologist, 1 geriatrician, 1 internist and 2 general practitioners) reviewed all clinical vignettes with the prescription support-tool (external validation) to confirm the likelihood of the clinical vignettes according to clinical practice and their readability.” However, the external validation seems to be more a consensus-based evaluation without any statistical analysis. In my opinion, there is a need for inter and intra-observer analysis to evaluate the reliability and reproducibility of the outcomes of the designed support-tool
Indeed, no statistical analysis was done, as the objective was not to assess the reliability of the tool, but to be sure that the clinical vignettes to be used in the randomized trial had a good clinical likelihood and readability. To make it clear, we deleted the term “external validation” in the clinical vignette paragraph.
Please consider revising the secondary outcomes wording (line 143-146) for a more precise text. I suggest “Secondary outcomes were: (1) appropriateness of each independent component of the primary outcome measure; (2) self-reported degree of confidence of physicians that their prescription of oral ATs complied with guidelines; and (3) self-reported usefulness of the prescription support-tool for physicians assigned to the experimental arm.
The modification was made (lines 157-159).
Please provide the p-values for table 1 where indicated. The group differences are not clear.
p-values were added in table 1 (page 11)
Round 2
Reviewer 1 Report
Authors considered with great attention my comments and questions. Their answers are fully motivated, and consequently the related modifications in the text are appropriate, and follow, when necessary, my suggestions.
I therefore think that the paper is now fully acceptable for publication.
Reviewer 2 Report
Thanks for your detailed responses and applied revisions.